# Mucus-Penetrating Silk Fibroin-Based Nanotherapeutics for Efficient Treatment of Ulcerative Colitis

**DOI:** 10.3390/biom12091263

**Published:** 2022-09-08

**Authors:** Dengchao Xie, Xin Zhou, Bo Xiao, Lian Duan, Zhenhua Zhu

**Affiliations:** 1Department of Gastroenterology, The First Affiliated Hospital of Nanchang University, Nanchang 330006, China; 2State Key Laboratory of Silkworm Genome Biology, College of Sericulture, Textile, and Biomass Sciences, Southwest University, Chongqing 400715, China; 3Chongqing Key Laboratory of Microsporidia Infection and Control, Southwest University, Chongqing 400715, China

**Keywords:** silk fibroin, resveratrol, nanoparticles, ulcerative colitis

## Abstract

Oral nanoparticles have been considered a prospective drug delivery carrier against ulcerative colitis (UC). To enhance the mucus-penetrating capacity and aqueous solubility, and strengthen the anti-inflammatory effect of resveratrol (RSV), we fabricated RSV-loaded silk fibroin-based nanoparticles with the functionalization of Pluronic F127 (PF-127). The obtained PF-127-functionalized RSV-loaded NPs had an average particle size around 170 nm, a narrow size distribution (polydispersity index < 0.2), and negative zeta potential (−20.5 mV). Our results indicated that the introduction of PF-127 strengthened the mucus-penetrating property of NPs. In vitro studies suggested that NPs with PF-127 enhanced the suppression of the secretion of proinflammatory cytokine TNF-α and reactive oxygen species (ROS) from RAW 264.7 macrophages under lipopolysaccharide stimulation in comparison with other counterparts. According to the evaluation of macro symptoms and main inflammatory cytokines, we further report preferable therapeutic outcomes achieved by PF-127 functionalized-NP-treated dextran sulphate sodium (DSS) groups in the colitis model compared with blank silk fibroin NPs and RSV-loaded NPs without the functionalization of PF-127. Taken together, this work suggests that the fabricated PF-127 NPs via the oral route are promising and useful RSV-loaded nanocarriers for UC treatment.

## 1. Introduction

Ulcerative colitis (UC), one of the most common chronic and incurable inflammatory bowel diseases (IBD), occurs along with the symptoms of diarrhea, blood in the stool, and mucus damage [1,2,3]. The increment in its incidence is allegedly due to the hardness of tackling genetic susceptibility, dysbacteriosis, excessive infiltration of intestinal inflammation, lesions of epithelial barriers, and destruction of redox equilibrium [4,5]. At present, the anti-ulcerative colitis drugs mainly include anti-inflammatory drugs, immunomodulators, corticosteroids, and biological agents, but long-term uptake will cause diarrhea, indigestion, and other side reactions [6,7]. It is arduous to meet the needs of efficient and safe modern clinical treatment for the sake of minimizing side effects.

Resveratrol (RSV), a natural organic compound of non-flavonoid polyphenol with anti-oxidation and anti-inflammatory properties, is found in many plants, such as grapes, peanuts, veratrum grandiflorum, etc. [8,9]. Certainly, it has favorable effects on the treatment of intestinal inflammation. Li et al. (2022) found that RSV can ease the condition by acting on enteric microorganisms and regulating critical and related signaling pathways (such as NF-κB) [10]. Maryam et al. (2015) pointed out that daily intake of 500 mg RSV can reduce the level of serum inflammatory markers (TNF-α) and moderate the severity of UC [11]. The high hydrophobicity, in vivo instability, and low oral bioavailability have impeded the further development and utilization of RSV [12]. Ongoing efforts to maintain and optimize the efficiency of RSV are urgently needed. Fortunately, with the organic integration of nanotechnology, nano-delivery possesses many advantages that can alleviate or even overcome some of the limitations of conventional drugs and natural compounds [13].

Among natural polymers, silk fibroin is worthy of attention. An increasing number of scientists are bracing for its modern potential in tissue engineering, biosensing, biomedical materials, and other fields [14,15]. As concerns drug delivery, as an FDA-approved polymer, silk fibroin extracted from the cocoon possesses good biocompatibility, biodegradability, modification-friendly properties, and is an excellent biomaterial for IBD drug delivery [16,17]. Lozano-Perez et al. (2014) demonstrated that the silk fibroin-based NPs optimized the controlled release of RSV, improved its stability during gastric passage and strengthened its bioactivity in the IBD model of rat colitis [9]. It is found that the therapeutic effect of silk fibroin-based nanoparticles with RSV was similar to result of dexamethasone, which is a classic drug used to treat IBD.

It is known that mucus, a slimy and viscoelastic gel, covers the surface of the colonic mucosa and has the ability to trap extraneous NPs with hydrophobic forces, hydrogen bonds, and electrostatic interactions to eliminate them from the gastrointestinal tract (GIT), thereby further suppressing the accumulation of NPs in the colonic inflammatory site [18,19,20]. It is essential for the actual treatment that orally administrated NPs possess the capability to cross the upper digestive tract and the mucus layer. Ideally, the NPs should ultimately gather in the corresponding inflammatory site. Pluronic F127 (PF127), an FDA-approved copolymer, is made up of one hydrophobic poly (propylene oxide) (PPO) component and two hydrophilic poly (ethylene glycol) (PEG) fractions [21]. Our proven research systemically dug into the colonic mucus penetration of NPs and showed that compared to PF127-free NPs, the PLGA NPs with surface functionalization of PF127 could optimize the penetrable properties of colonic mucus [22,23].

Nevertheless, to our knowledge, there have been no attempts taken to research the influence of combining Pluronic F127, silk fibroin, and RSV against UC. In this work, we made an investigation with PF127-modified RSV-loaded silk fibroin-based nanoparticles (PF127-RSV-NPs) to understand its physicochemical properties, its anti-inflammatory activities, and its antioxidant activities in vitro, as well as to subsequently evaluate its therapeutic outcomes against UC in mice.

## 2. Materials and Methods

### 2.1. Materials

Silkworm cocoons were obtained from the State Key Laboratory of Silkworm Genome Biology, Southwest University (Chongqing, China). Pluronic F127 (PF127), camptothecin (CPT), hydroxyethyl cellulose, and lipopolysaccharide (LPS) were obtained from Sigma-Aldrich (St. Louis, United States). Resveratrol, dimethyl sulfoxide (DMSO), alginate and chitosan were obtained from Aladdin (Shanghai, China). The myeloperoxidase (MPO) kit was obtained from Nanjing Jiancheng Bioengineering Institute (Nanjing, China). The enzyme-linked immunosorbent assay (ELISA) kits for the quantification of inflammatory cytokines (TNF-α, IL-6, IL-1β) were provided by Solarbio (Beijing, China). The hematoxylin and eosin (H & E) staining kit was obtained from the Beyotime Institute of Biotechnology (Nanjing, China). Dextran sulphate sodium (DSS, 36–50 kDa) was obtained from MP Biomedical Inc. (Aurora, CO, USA).

### 2.2. Extraction and Purification of Regenerated Silk Fibroin

Silkworm cocoons in the absence of the silkworm chrysalis were torn into pieces, boiled twice in the Na_2_CO_3_ aqueous solution (0.5%, *w*/*v*), then rinsed with deionized water to adequately wipe off the outermost sericin proteins. Followed by complete drying, the cocoons were dispersed in the classic Ajisawa’s reagent made up of calcium chloride, anhydrous ethanol, and water (1:2:8, the molar ratio) at 78 °C for 2 h (averting boiling). Irrelevant substances were eliminated to improve the purity by constant-temperature centrifugation for 10 min at 8300× *g*. As for the supernatant, the acquisition was further purified by the dialysis method with deionized water for three days to remove the residual CaCl_2_ and other impurities via a cellulose semipermeable membrane (MWCO = 1400 Da). The final regenerated silk fibroin (RSF) was placed at 4 °C after filtration and freeze-drying.

### 2.3. Preparation of Various NPs

The RSV-loaded silk fibroin-based nanoparticle with PF-127 was obtained using a miscible organic solvent technique by following prior studies [24]. In brief, RSF was dissolved in deionized water (10 mg/mL). Then, PF-127 was added into RSF solution. The mass ratios of PF-127 to dry weight RSF were set as 1% and 5%. Meanwhile, the RSV solution was prepared by dissolving RSV in cold acetone (10%, *w*/*v*). Next, the PF-127/RSF solution was dropped into the RSV solution with gentle stirring. The emulsion was formed instantaneously with vorticity for 30 s, followed by the ultrasonic treatment at an amplitude of 30% for 2 min in the 0 ℃ environment. It was stirred adequately in the ventilation installation to vaporize acetone. The final NPs were acquired through centrifugation at 13,000× *g* for 20 min, rinsed, and lyophilized before storage at −20 °C. The steps of the fabrication of RSV-NPs without PF-127, except for the mixture of RSF and PF-127, were the same, as follows.

### 2.4. Physicochemical Characterization of NPs

The parameters about particle size and zeta potential were measured via a dynamic light scattering technique (Malvern Zeta Sizer NanoS90, Malvern, UK). To evaluate the stability of NPs, various NPs were incubated in solutions with different pH values (7.4 and 6.8) at 37 ℃ to simulate intestinal and colonic fluid, and simultaneously shaken at 150 rpm. The change of particle sizes and zeta potentials were observed by a dynamic light scattering instrument. The morphologies of NPs were observed using transmission electron microscopy (TEM, LEO 906E, Zeiss, Jena, Germany). The quantification of the loaded drug in silk fibroin NPs was measured according to the previous method [17,24,25]. Briefly, NPs were dissolved in DMSO. Then, the obtained solution was transferred into a 96-well plate to measure the absorbance at 320 nm using a multimode plate reader (PerkinElmer, Boston, MA, USA). The amount of loaded RSV was obtained from the standard curve of RSV. The drug loading and encapsulation efficiency of RSV-loaded NPs were calculated according to Equations (1) and (2):(1)$$\mathrm{Drug}\;\mathrm{loading}\;\left(\%\right)=\frac{Weight\;of\;drug\;in\;RSV-NPs}{Weight\;of\;RSV-NPs}\times 100\;\%$$
(2)$$\mathrm{Encapsulation}\;\mathrm{efficiency}\left(\%\right)=\frac{Weight\;of\;drug\;in\;RSV-NPs}{Weight\;of\;input\;drug\;}\times 100\;\%$$

The *X*-ray diffraction (XRD) spectra of relative samples were detected using scanning from 10° to 50° and operated at 40 kV and 30 mA at the velocity of two degrees per minute (XRD-7000, Shimadzu, Japan). NPs were tested via a circular dichroism (CD) spectrometer (Bio-Logic, Seyssinet-Pariset, France) to track secondary structures ranging from 190 to 250 nm.

### 2.5. Mucus-Penetrating Evaluation of NPs

The mucus penetration analysis of NPs was made by referencing our prior method [26]. Hydroxyethyl cellulose (HEC) aqueous solution (0.2% *w*/*v*) was prepared by overnight magnetic stirring as a mucus-simulating hydrogel. Next, the dried NPs were resuspended in the buffer at pH 6.8 (0.1%, *w*/*v*). A drop of the obtained NPs suspensions was squeezed out into the Petri dish with a pipette (Thermo Fisher, Waltham, MA, USA). The tracks of NPs in a mucus-simulating atmosphere were investigated via a CCD camera attached to a fluorescence microscope (Olympus IX83, Tokyo, Japan) with a frame rate of 30 fps for 2 s. The mean-square-displacement (MSD) values were gained from the track data.

### 2.6. Drug Release Profiles of NPs

The dialysis method to investigate the controlled release behavior of these NPs was adopted, as depicted in our previous studies [25,27]. NPs containing an equivalent drug Camptothecin (CPT) of 200 μg were suspended in deionized water, regardless of whether PF-127 was functionalized or not. Then the suspensions were shifted into dialysis bags (MWCO = 1400 Da). The two sections of sealed bags were immersed in 50 mL centrifuge tubes containing the corresponding media (acidity, H_2_O_2_), which were placed at 37 ℃ for shaking at 150 rpm. In the meantime, Tween-80 (0.1%, *w*/*v*) was subjoined moderately into every medium to retain the water solubility of water-insoluble CPT. At designated time intervals, the contents of drugs released from NPs were monitored in the corresponding medium using a plate reader (Bio-Tek Instruments, Winooski, VT, USA) at 360 nm excitation wavelength and 430 nm emission wavelength.

### 2.7. Bioactivity of NPs in Cells

RAW 264.7 macrophages were incubated in 96-well plates at a density of 2 × 10^4^ cells per well with Dulbecco’s modified Eagle’s medium containing fetal bovine serum at 37 °C, 5% CO_2_. The used culture medium was removed and rinsed with cold PBS after complete overnight incubation. Fresh media containing various NPs with different concentrations were supplemented to each well later. After co-incubation with various NPs, cells were incubated with MTT for 4 h. The MTT concentration was 0.5 mg/mL. The viability of RAW 264.7 macrophages was measured using the absorbance at 570 nm by a multifunctional microplate reader (Crailsheim, Germany).

### 2.8. In Vitro Anti-Inflammatory Activities of NPs

The expression of inflammatory cytokines was quantified by the ELISA kits. The RAW 264.7 macrophages suspension was dispensed into a 96-well plate at the density of 2 × 10^5^ cells per well. Fresh media containing various NPs (equivalent to 2–8 μM of RSV) were used to replace the cell culture media and added to each well. After incubation for 24 h in the dark, the cells were exposed to LPS (500 ng/mL) for 3.5 h. Following the collection of supernatants, each group’s concentration of TNF-α was quantified. Cells treated with LPS only were denoted as a positive control.

### 2.9. In Vitro Antioxidant Activity of NPs

The experiment was divided into the test of intracellular ROS scavenging capacity and the classical DPPH assay. The suspension of macrophages (2 × 10^5^ cell per well) was planted into a 24-well plate and after incubation overnight, exposed to various NPs immersed in the serum-free medium (equivalent RSV concentration: 8 μM). The total weight of blank silk fibroin NPs was the same as those NPs containing drugs. Each well was subjoined with complete culture medium after 4.5 h incubation. Except for the negative control, each group was treated with LPS (500 ng/mL) for 3.5 h. Next the cells were stained with DCFH-DA (10 μM) for 20 min in lightless condition at room temperature in accordance with the relevant kit operation. Cells stained with DCFH-DA were measured using flow cytometry (FCM, ACEA Novocyte™, USA). The negative control group was not stimulated with LPS and NPs, whereas the group just stimulated by LPS was treated as a positive control.

The evaluation of NPs towards DPPH free radical molecules was made by referencing a previous method [28]. In short, different dried nanoparticles were dispersed in deionized water with different NPs concentrations ranging from 0.1 to 2 mg/mL. Then, the suspensions were brought into DPPH solution and absolute ethyl alcohol at the same volume. The mixture was incubated for 30 min in a dark place. The equal DPPH solution was stirred in the absence of NPs as reference. The radical scavenging activity of NPs was evaluated by measuring the absorbance around 520 nm of groups above in UV–Vis absorption spectra by a multifunctional microplate reader (Crailsheim, Germany).

### 2.10. In Vivo Suppress Effects of NPs against UC

This in vivo study was executed on the basis of the Guidelines for Care and Use of Laboratory Animals of Southwest University and the protocols were approved by the Animal Ethics Committee of Southwest University. Female FVB mice were divided into 6 groups at random: the healthy control group (given with water only), the DSS control group, the Blank silk fibroin NPs-treated DSS group (SF-NPs group), the RSV-NPs-treated DSS group (0% PF-127 group), the 1%PF-127-RSV-NPs-treated DSS group (1% PF-127 group), and the 5%PF-127-RSV-NPs-treated DSS group (5%PF-127 group). To protect corresponding NPs through the upper GIT and gain colon-targeted drug delivery efficiently, mice were orally administrated with NPs (5 mg RSV per kg mice) each day with a quantitative formation of hydrogels that consisted of chitosan and alginate. The weight changes of the groups were recorded daily. All mice were euthanized on the eighth day. Subsequently, the colons were gathered for further research. Additionally, MPO activities in the colon were quantified using the MPO kit. The contents of the inflammatory cytokines in the serum were measured using relevant kits.

### 2.11. Statistical Analysis

Statistical analysis was conducted using Student’s *t*-test or Duncan’s multiple-range test to analyze the significant differences of mean values. Results represented mean ± standard error of the mean (S.E.M). * *p* < 0.05 and ** *p* < 0.01 were applied to express statistical significance.

## 3. Results and Discussions

### 3.1. Physicochemical Characterization of NPs

The formation of silk fibroin NPs was examined using the water-miscible organic solvent method that facilitates smaller sizes than do other methods, such as phase separation in salt and freezing [29,30]. Acetone was used as the RSV solvent to fabricate the NPs as previously reported [31]. Numerous studies suggested that the particle size and zeta potential were closely related to the effect of the drug release, in vivo stability, and other factors [21,32]. The dynamic light scattering (DLS) data of the fabricated NPs are presented in Table 1. The mean size of PF-127-NPs (1% PF-127), PF-127-NPs (5% PF-127), and RSV-NPs was around 150–170 nm. The zeta potential of NPs ranged from −27.2 to −19.4 mV. NPs were with narrow distribution (PDI < 0.2) and sphere via TEM (Figure 1a–d). It should be emphasized that the size discrepancy by DLS and TEM was credited with the test environment and the difference in pretreatment, as we had discovered before [31,33].

The crystallinity of hydrophobic drugs in NPs should be determined by the factor that has a bearing on drug release [34]. To assess molecular interactions between the pristine RSV and the silk fibroin matrix, X-ray diffraction (XRD) patterns of pristine RSV, blank silk fibroin-based NPs, RSV-NPs, and PF-127-NPs were determined (Figure 1f). We found that the pure RSV showed several dense and sharp peaks in the XRD patterns, implying the intrinsic crystalline structure mode. Additionally, all samples in the form of NPs had a curve with considerable smoothness and no obvious peak. The results reflected that the crystallization process of RSV was obstructed by SF, and RSV formed an amorphous state with the matrix molecules.

With circular dichroism (CD), our current studies reported that the solutions of regenerated SF molecules had a feeble negative peak centered at around 210 nm, implying that its random coil conformation in an aqueous solution had no stable secondary structure [25,35]. As is shown in Figure 1e, all silk fibroin-based NPs showed a positive peak around 200 nm along with the emergence of a negative peak at 220 nm. This result was consistent with our previous study that when RSF changed to NPs, a conformational change from random coils to β-sheet configuration occurred [24].

### 3.2. Mucus-Penetrating Capacities of NPs

Oral administration was one of the most common routes for IBD therapy. The harsh condition in the GIT such as pH variation was considered a rate-limiting step for the research and development of medicament by oral route [13]. It is necessary to explore the stability of NPs in in vivo simulated environment by evaluating the size and zeta potential prior to the mucus-penetrating capacities of NPs. As indicated in Figure 2a, there was no dramatic change in particle size of NPs regardless of whether PF-127 was modified or not during the incubation in simulated small intestinal fluid, so was the change of zeta potential. The zeta potentials of three types of NPs basically kept stable (Figure 2b) in simulated colonic fluid, but interestingly, the size of NPs without PF-127 increased sharply, almost 2.5 times as higher as the initial one. In the meantime, the sizes of PF-127-NPs (both 1% and 5% PF-127) showed merely a slight decrease with the time going by. Hereto, this consequence might be due to the explanation that the hydrophilic segment (PEO) of PF127 tended to stretch outward and prevent the aggregation of NPs in the form of a hydrophilic outer layer, whereas the hydrophobic sections (PPO) of PF127 assembled with the insoluble secondary structures of silk fibroin-based NPs to improve their stability [36,37].

As we introduced above, although the drug can be stabilized by special hydrogels and silk fibroin materials with PF-127, mucus at the surface of the colonic mucosa can trap NPs and impede the accumulation of NPs in the colonic inflammatory tissue, in turn influencing the final effect beyond a shadow of doubt. To evaluate the capacity of NPs, the HEC hydrogel was chosen as a mucus-simulating gel, which has similar microrheology characteristics and chemical composition to mucus [23]. The movement of RSV-NPs suffered from the obstacle as depicted in Figure 2c, showing a relatively overlapping track in the mucus-simulating hydrogel, whereas NPs with PF-127 presented relatively free diffusion. We also examined variations of MSD, which were considered a significant index to evaluate the locomotive behavior of NPs in mucus-simulating gel [26]. It was found that the PF-127-NPs (5%) ranked first in the moving ability, confirmed by the MSD result (Figure 2d). Our results suggest that the introduction of PF127 helped motivate the osmosis and migration of NPs across the muco-environment. PF-127 on the surface of NPs tended to reduce the physicochemical forces between mucin glycoproteins and NPs [38], which may be responsible for the improvement of NPs with PF-127 in mucus.

### 3.3. Drug Release Profiles of NPs

The controlled release of drugs from NPs is an important property for the efficient establishment of the oral drug delivery system [25]. Given the interaction and oxidization of RSV in oxidation environment (H_2_O_2_), CPT replaced RSV as the test drug. Figure 3a showed that nearly 60% of the encapsulated drugs were released from NPs in the buffers at pH7.4 during the 120-h incubation in a time-dependent manner. However, the drug release profile in the buffer at pH 5.5 was higher than that in the buffer at pH 6.8 (68.09%), which was similar to the previous findings [24,31]. In more detail, the three curves show a phenomenon where, during the initial 24 h incubation, each NP displayed rapid release followed by a slower release rate. The diffusion of the drugs in the superficial layer of the NPs accounted for the initial burst drug release profile. The drugs released from the interior of the NPs caused a subsequent relatively flat curve [34]. In addition, we found that the cumulative release proportion for PF-127-NPs (5% PF-127) was almost the highest among the groups at all points-in-time at pH7.4 (Figure 3b) and 6.8 (Figure 3c). It was speculated that compared with others, more drugs from the surface layer of PF-127-NPs (5% PF-127) were released into the solution in these environments. Perhaps this was related to the enhancement of matrix swelling in this environment.

Taking into account the overproduction of ROS in the inflamed colon [24], it is rational to speculate that NPs with PF-127 had obvious H_2_O_2_-responsive capacity. Herein, NPs were incubated in the buffer at pH 6.8 with H_2_O_2_ (10 mM) to verify the accelerated release effect at the simulated colonic fluid with ROS production. The result shown in Figure 3d was that 95.52% of CPT was released from PF-127-NPs (5% PF-127) after 5 days in the incubation environment of 10 mM H_2_O_2_ at pH 6.8, and the drug release from PF-127-NPs (1%) also had been accelerated, which was significantly higher than the group without the addition of H_2_O_2_ (71.1%).

The researches show that the PLGA (poly (lactic-co-glycolic acid))-based protein tissue engineering scaffold with the PF-127 can accelerate the release of protein compared to the scaffold without PF127 [36,37]. The hydrophilic poly (ethylene oxide)-chain segments of Pluronic may tend to be segregated onto the surface of the carrier, thus regulating the hydrophilicity of the resulting material and promoting drug diffusion [39]. Certainly, the hypothesis will need to be further investigated in the future.

### 3.4. In Vitro Anti-Inflammatory Activities of NPs

As non-negligible cells are closely related to the research of UC, RAW 264.7 macrophages were chosen under investigation, as follows [24]. At first, the in vitro cytotoxicity of fabricated NPs was performed by the MTT assay in RAW 264.7 macrophages. It can be seen from Appendix A that no matter if PF-127 was modified, there was no obvious cytotoxicity (viability more than 80%). Pro-inflammatory cytokines are highly overexpressed in the progression of UC, which leads to tissue damage [40]. Inhibition of pro-inflammatory cytokines tumor necrosis factor-alpha (TNF-α) produced by macrophages has been a common therapy strategy in IBD, as shown in clinical trials using the monoclonal antibody and infliximab [41]. As shown in Figure 4, LPS obviously enhanced the TNF-α level when compared to the negative control. To that point, we found that RSV-NPs were capable of enhancing the suppression of TNF-α when conjugated with PF-127 and the NPs showed a dose-dependent increment in the inhibition of TNF-α. Significant differences in the level of TNF were witnessed between PF-127-NPs and the RSV-NPs without PF-127. Moreover, to some extent, we noted that blank NPs had the capacity to suppress the expression of TNF-α due to the intrinsic anti-inflammatory ability of silk fibroin, as reported previously [31,42]. Focusing on the content of our previous group, Zhou et al. (2019) designed the curcumin (CUR)-loaded PLGA-NPs with PF127, which could greatly intensify the mucus penetration of NPs and the treatment effect in UC model mice [22]. Additionally, we found that oral nanotherapeutics based on porous PF127-functionalized CUR-loaded NPs also showed a greater therapeutic outcome against UC [34].

### 3.5. Anti-Oxidant Activities of NPs

Reactive oxygen species (ROS), including hydroxyl radical, superoxide anion, and hydrogen peroxide, is of importance for mucosal injury in gastrointestinal disorders [43]. Although the precise molecular pathways of colon cell damage are uncovered, recent studies demonstrated that excessive ROS generated in the inflamed colon may drive uncontrolled oxidative stress, thus causing oxidative damage to nucleic acids, proteins, and lipids [44]. To determine quantitative feedback, we performed an analysis by FCM. The FCM histograms exhibited the same trends as observed in Figure 5a, which revealed that the peaks of all the NPs-treated groups relative to the positive control were shifted to the left, indicating the antioxidant effect of NPs. Compared to the positive control, the ROS-related signal was lower after the administration of RSV-NPs, which was also confirmed via the mean fluorescence intensity (MFI) (Figure 5b).

Additionally, the antioxidant capacity of NPs by DPPH was assessed in a concentration-dependent manner in Figure 5c. The scavenging effect rose with the increase of concentration for each group and the enhanced scavenging effect was presented on PF-127 NPs (1% PF-127). But no significant alteration was found between PF-127 NPs and non-functionalized NPs loaded with RSV in the absence of cell interaction. It is worth noting that silk fibroin with the inherent antioxidant property also enhanced the efficacy of RSV via a synergistic effect. From the perspective of its structure, it was perhaps ascribed to 4% of aromatic residues in silk fibroin, which present the radical scavenging activity [45].

### 3.6. In Vitro Therapeutic Efficacies of NPs against UC

Some symptoms of DSS-induced UC mice are similar to those in UC human patients, such as the apparent decline of body weight, the shortening of colonic length, the destruction of colon epithelium barriers, and the infiltration of inflammatory cells [23]. To comparatively evaluate the therapeutic activity of NPs via oral administration against UC, we utilized a mouse model with 3.5% DSS and further treated the mice with different NPs. In addition, it is known that drug delivery is required in an appropriate way in UC therapy. The relationship between the delivery route and the therapeutic effect is intimate [13]. Though oral administration suffers from some problems, as explained above, it remains an attractive option because it not only possesses the advantages of high patient compliance, non-invasiveness, and cost-effectiveness, but also provides convenient drug delivery to the colonic mucosal layer [46]. In order to guard the loaded drug from degradation in the upper GIT and make the drug stimuli-responsive upon release when the hydrogel was broken up in the colon lumen, UC mice were treated with NPs embedded in chitosan-alginate hydrogels via gavage administration [24,47]. Figure 6a depicted that the bodyweight of the healthy control group showed a steady and moderate increase throughout the whole course, whereas the body weight of the DSS control group had an obvious reduction. To our relief, all NPs can more or less mitigate the bodyweight loss with the treatment of DSS, especially for NPs with 1% PF-127 (a mere 5.3% decrease in overall weight). Owing to several factors like the disruption of the epithelial barrier and the mucosal lesion, the colon length becomes a key visual indicator of inflammation in the correlational UC research [22]. In Figure 6b, it is noted that the PF-127-NPs-treated groups represented longer colon length than the DSS control group and the differences were statistically significant (*p* < 0.01). Especially, 1% of PF127-RSV-NP-treated DSS group ranked first (7.2 cm) among these NP-treated DSS groups in terms of colon length. We also found that the change in spleen weight was correlated with substantial outcomes of weight loss and colon shortening (Figure 6c). The spleen weight of the PF127-RSV-NP-treated group came extremely close to that of the healthy control group. The significant difference was indicated between the 1%PF127-RSV-NP-treated DSS group and the DSS control group (*p* < 0.01).

MPO, a typical endogenous enzyme, was mainly produced by neutrophil granulocyte. Compared with the high expression in terms of the MPO activity in the DSS control group, as shown in Figure 6d, there is a certain decrease for the treatment of blank NPs, RSV-NPs, and PF-127-NPs, respectively. Some evidence has manifested that the overexpression of TNF-α, IL-6, and IL-β were implicated in the deterioration of UC [48]. Figure 6e presents that in comparison with the healthy control group, the concentrations of TNF-a, IL-6, and IL-1β for the DSS control group were significantly higher than any others. However, oral hydrogel-embedding NPs had the capacity to inhibit the expression levels of these pro-inflammatory cytokines. Blank NPs and RSV-NPs reflected relatively poor inhibition of pro-inflammatory cytokine levels. The results were consistent with the consequences in vitro above.

The histological changes in colon tissue sections were stained with H & E to analyze the morphology treated with different NPs. Figure 7a revealed that the DSS-control group disrupted the epithelial barrier, crypt, and gathering of immune cells. As expected, the hydrogel-embedding PF127-RSV-NPs reversed the severe depletion of goblet cells and the occurrence with massive infiltration by immune cells and showed mild damage in the epithelial layer and clear decreases of inflammatory cell infiltration in the mucosa. Meanwhile, the histological score was measured (Appendix A). Around 1% PF127-RSV-NPs had a relatively low score, suggesting the best therapeutic effect with oral administration among counterparts.

Attention is being focused on the tight junction, which is a key role in maintaining the homeostasis of the colonic epithelial barriers [49]. Clinical studies demonstrated that the quantities of tight junction proteins abnormally decreased in the colon tissues of UC patients [50]. We evaluated the expression of zonula occluded-1 (ZO-1), which was one of the primary tight-junction proteins. As seen in Figure 7b, the positive region of the PF-127-NPs-treated group (1% PF-127) was larger than that of the DSS control group and the other NPs-treatment groups. Together with the analysis of the expression profile of ZO-1 (Appendix A), the results revealed that peroral PF-127-NPs can relieve the destruction of the colonic epithelial barrier.

Taken all together, given the evaluation results of the macroscopic symptoms and inflammatory markers, the in vivo experiments showed that the moderate PF-127 with grafted silk fibroin NPs loaded with RSV had a better effect than other treatments, likely leading to a substantially enhanced therapeutic efficacy for the loaded drug for the in vivo model, thus becoming an attractive and promising strategy for IBD treatment.

## 4. Conclusions

In this study, we fabricated PF-127-grafted silk fibroin-based NPs. The PF-127 NPs had a desirable average size (around 250 nm) and negative surface charge (around −20 mv). Additionally, PF127-NPs presented desirable capacity for anti-inflammatory and antioxidant activity in vitro and evidently enhanced the mucus penetration of NPs. In vivo experiments revealed that hydrogel-embedding functionalized NPs achieved a better therapeutic effect against UC than other counterparts. Collectively, Pluronic F127-grafted silk fibroin-based NPs could be exploited as an efficient carrier for UC therapy. These findings are valuable for optimizing the biological activities of natural compounds with beneficial properties for human health and wellness.

## Figures and Tables

**Figure 1 biomolecules-12-01263-f001:**
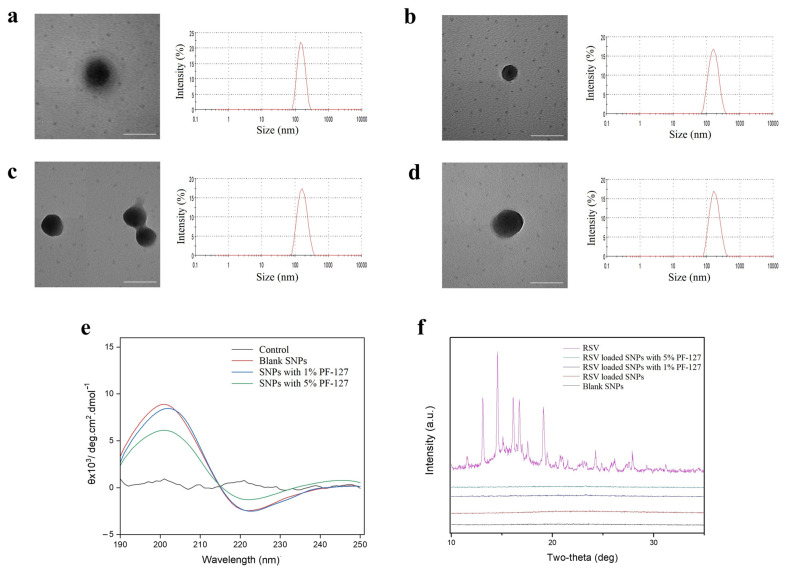
Physicochemical characterization of NPs. Representative TEM images and the corresponding size distribution profiles of (**a**) blank NPs, (**b**) RSV-NPs, (**c**) PF-127-RSV-NPs (1%), and (**d**) PF-127-RSV-NPs (5%). Scale bar represents 200 nm. (**e**) CD spectra and (**f**) XRD patterns of pure resveratrol, blank silk fibroin-based NPs, PF-127-RSV-NPs (1%), and PF-127-RSV-NPs (5%).

**Figure 2 biomolecules-12-01263-f002:**
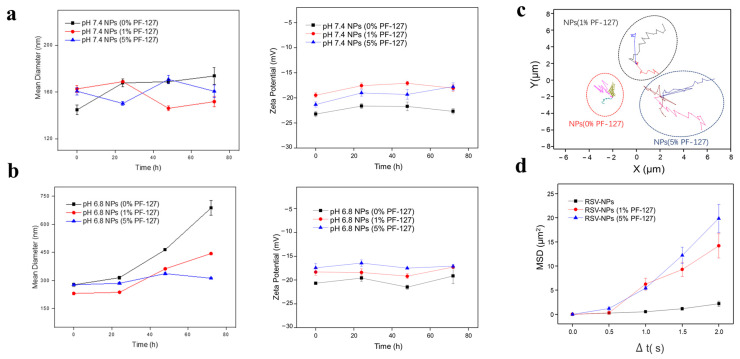
Stability and mucus penetration of NPs. Variations of hydrodynamic diameters and zeta potential of NPs during incubation in simulated intestinal fluid (**a**) and simulated colonic fluid (**b**) for 3 days. (**c**) Representative motion trajectories and (**d**) average MSD of various NPs in the mucus-simulating hydrogel. Data are expressed as means ± S.E.M, *n* = 3.

**Figure 3 biomolecules-12-01263-f003:**
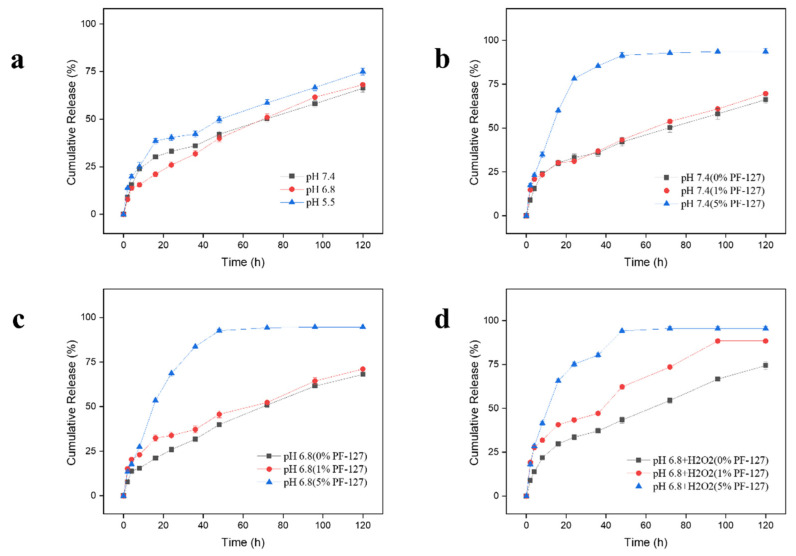
In vitro cumulative release profiles of drugs from undecorated NPs in buffers with (**a**) different pH values. In vitro drug release behaviors of NPs in buffers with pH (**b**) 7.4, (**c**) 6.8, and (**d**) 6.8 values and 10 mM H_2_O_2_. Data are expressed as means ± SD (*n* = 3).

**Figure 4 biomolecules-12-01263-f004:**
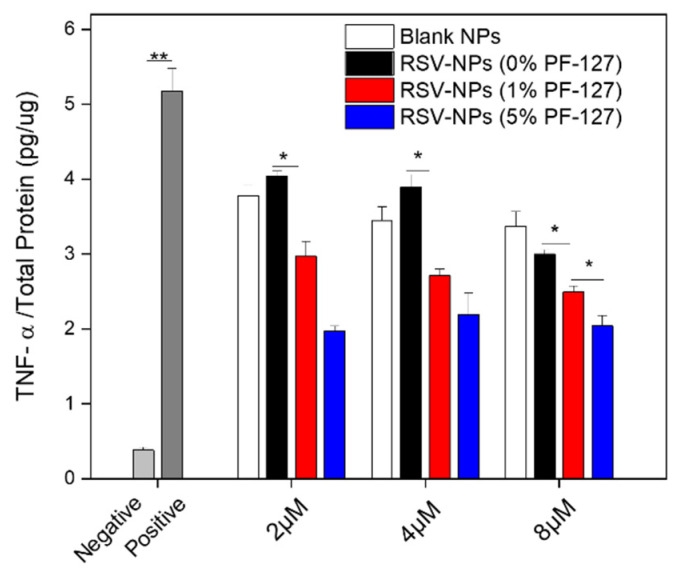
In vitro anti-inflammatory activities of blank NPs, RSV-NPs, PF-127 NPs (1%), and PF-127 NPs (5%) against RAW 264.7 macrophages. The amounts of TNF-α were quantified by using the ELISA method. Each point represents the mean ± S.E.M (*n* = 3; * *p* < 0.05 and ** *p* < 0.01, Student’s *t*-test).

**Figure 5 biomolecules-12-01263-f005:**
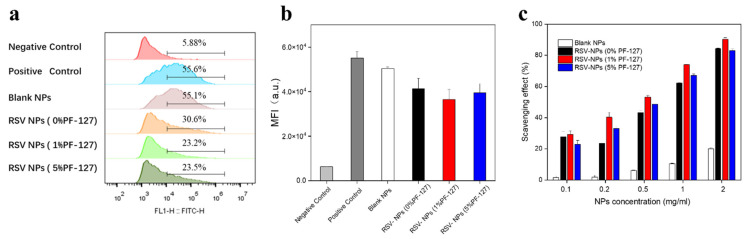
In vitro antioxidant activities of NPs. (**a**) Flow cytometric histograms of RAW 264.7 macrophages determined by DCFH-DA staining after treatment of various NPs. (**b**) MFIs of RAW 264.7 macrophages treated with various NPs showing intracellular ROS signals. (**c**) The radical scavenging activity of various NPs by the DPPH assay. Each point represents the mean ± S.E.M. (*n* = 3).

**Figure 6 biomolecules-12-01263-f006:**
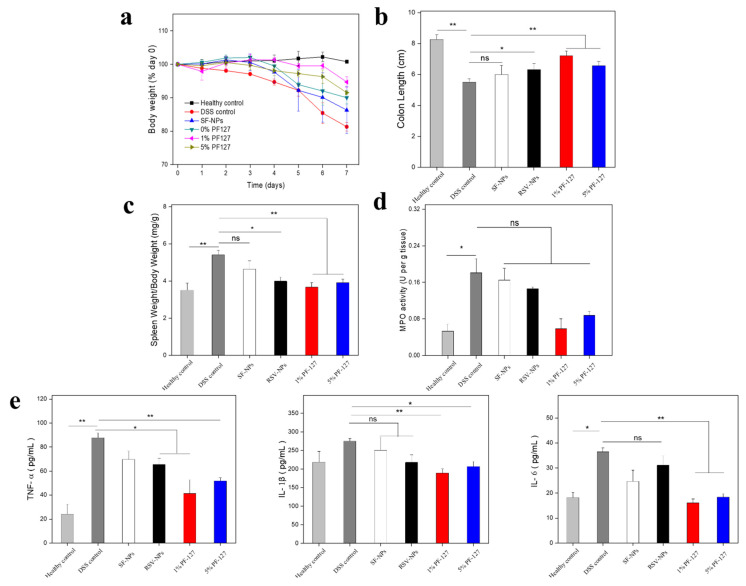
In vivo therapeutic effect of NPs against UC via oral administration. (**a**) Time-dependent variations of body weight over time, normalized to the percentage of the day-zero body weight. (**b**) colon length, (**c**) spleen weight, and (**d**) colonic MPO activity in different mice. (**e**) TNF-α, IL-1β, and IL-6 concentrations in serum. Each point represents the mean ± S.E.M. (*n* = 5; * *p* < 0.05, ** *p* < 0.01, and ns = no significance).

**Figure 7 biomolecules-12-01263-f007:**
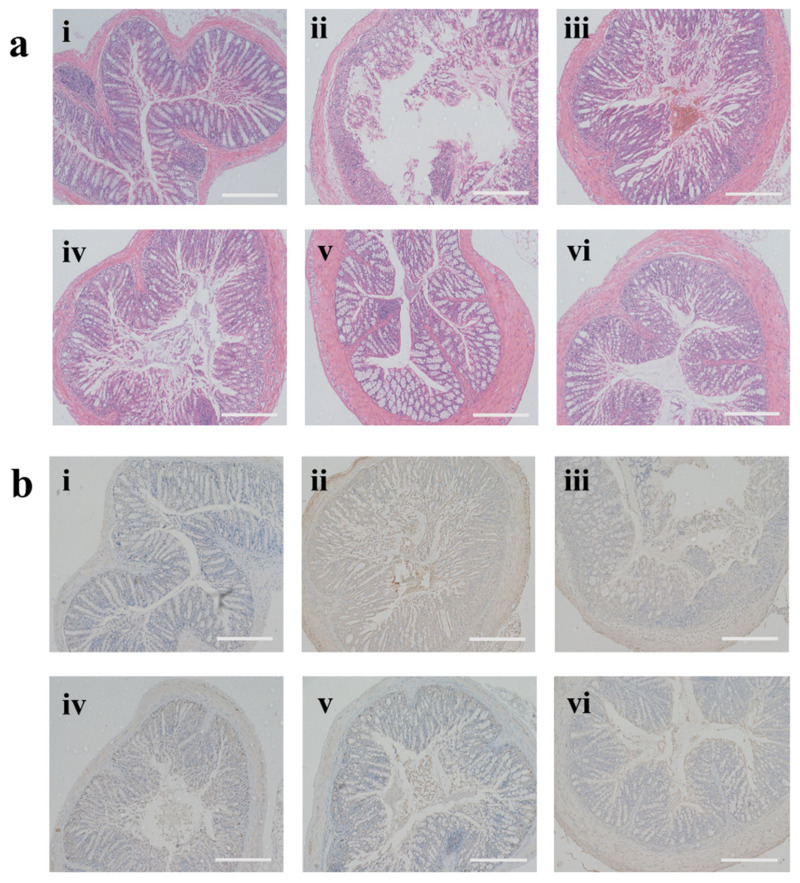
(**a**) H & E-stained colon tissues and (**b**) the expression profiles of ZO-1 in different mouse groups. (i) healthy control, (ii) DSS control, (iii) SF-NPs-treated DSS group, (iv) RSV-NPs-treated DSS group, (v) PF-127-NPs-treated DSS group (1%), and (vi) PF-127-NPs-treated DSS group (5%). Scale bar represents 200 μm.

**Table 1 biomolecules-12-01263-t001:** Characteristic of NPs (mean ± S.E.M., *n* = 3).

Name	Particle Size(nm)	PDI	Zeta Potential(mV)	Drug Loading (%)	Encapsulation Efficiency (%)
Blank NPs	155.8 + 1.2	0.183	−28.3 + 0.4	/	/
RSV-NPs	153.8 + 0.6	0.081	−25.2 + 0.3	5.8%	40.6%
RSV-NPs(1% PF-127)	160.3 + 1.6	0.104	−19.9 + 0.5	2.3%	31.5%
RSV-NPs(5% PF-127)	168.1 + 2.2	0.101	−21.2 + 0.5	4.7%	44.8%

## Data Availability

Not applicable.

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
