# Peer review of "Mucus-Penetrating Silk Fibroin-Based Nanotherapeutics for Efficient Treatment of Ulcerative Colitis"

_biomolecules, 2022, doi:10.3390/biom12091263_

Round 1
Reviewer 1 Report
The manuscript entitled “Mucus-penetrating silk fibroin-based nanotherapeutics for efficient treatment of ulcerative colitis” presents the functionalization of resveratrol (RSV) loaded silk fibroin nanoparticles (NPs) with Pluronic F127 (PF127) and their potential use for the treatment of ulcerative colitis (UC). The hypothesis of the study is that PF127 functionalization will enhance the RSV loaded NPs to the intestinal epithelium and thus provide a more effective treatment. RSV is a promising candidate of natural origin for the prevention and amelioration of intestinal inflammation and both the lipopolysaccharide stimulated RAW 264.7 macrophages and the dextran sodium sulfate induced colitis model that are used in this study are well established models of the UC. The authors have published a few studies on the potential use of drug loaded NPs for the treatment of UC. In terms of originality, there are very few studies where silk fibroin nanoparticles have been loaded with resveratrol for the treatment of ulcerative colitis, but in none of them Pluronic F127 has been used to enhance their mucus-penetrating ability.
Nevertheless, I recommend to reconsider the acceptance of the manuscript after major revision, because of the plethora of corrections that must be made (some of them are serious), not to mention the problems that arise from the poor use of the English language. Please find below my comments.
Abstract
Line 16: After the word resveratrol the abbreviation (RSV) should be added, since this is the first time the word appears in the abstract.
Line 19: According to Table 1 in the Results section the z-potential of the PF127 functionalized NPs is ~ -20,5 and not -23 mV.
Line 25: Before the abbreviation DSS the word dextran sodium sulfate should be added, since this is the first time the abbreviation appears in the abstract.
Introduction
Line 44: The year of publication (2022) should be added after the reference Li et al..
Line 46: The reference Maryam et al. does not match with the reference #11 in the reference list. Also, the year of publication should be added after the reference.
Line 48: Probably “in vivo” is meant instead of simply “vivo”.
Line 58: The reference J Galvez et al. slould be changed to Lozano-Perez et al. and the year of publication (2014) should be added as well.
Materials and Methods
Section 2.1: The source of Hydroxyethyl cellulose is not mentioned.
Lines 119 - 121: It is somehow unclear how the quantification of RSV in the NPs was done. First, the authors claim that DMSO was used to wash away the drugs from the NPs. It is not well justified why this step was taken. If it was to remove the absorded drug on the surface of the NPS, then the authors underestimate the total amount of RSV carried by the NPs, because nowhere else in the manuscript it is mentioned that the DMSO wash proceeded the other experiments. Then the authors claim that the NPs were transferred in a 96-well plate and the amount of RSV was quantified with the use of a RSV standard curve. Usually, standard curves are acquired with known amounts of the pure substance. One would expect that the authors should have first somehow dissolved the NPs in order to acquire the loaded RSV and then quantify it with a standard curve.
Section 2.6: The description of the experiment is poor: The equivalent drug is not mentioned, it is not clear what “the two sections of sealed bags” are and what the corresponding media are. Finally, it is not clear why a dialysis bag was used. One would expect that the NPs would be directly suspended in the release medium.
Lines 178 - 180: It is unclear what the authors mean by the last sentence.
Line 190: Probably “in vivo” is meant instead of simply “vivo”.
The experiment for the evaluation of NPs stability in intestinal and colonic fluid (see lines 240 - 254 in the Results section) is not described in the Materials and Methods section.
No statistical methods for interpretating data are mentioned in the Materials and Methods section.
Results
Line 212: Probably “in vivo” is meant instead of simply “vivo”.
Lines 253 - 254: The PPO section is hydrophobic, so the argument needs rephrasing.
Line 263: HEC instead of EC should be written.
Lines 279 - 281: It is not absolutely true that “NPs had a pH-Stimulus responsiveness”, because, although the release profile at pH 5.5 was higher than at pH 6.8, according to Figure 3a, the release profile at pH 7.4 was also higher than at pH 6.8.
Line 305: The year of publication should be added after the reference Lee et al..
Line 307: The reference is wrong and thus the claim can not be verified.
Line 314 - 316: Figure S1 is not provided, so the claim can not be verified.
Lines 327: The year of publication should be added after the reference Zhou et al..
Lines 327 - 329: The reference Zhou et al. does not match with reference #42.
Figure 4 and lines 321 - 327: The interpretation of results of Figure 4 raises a serious doubt if the anti-inflammatory activity are really attributed to RSV or the PF127 itself, since NPs with no RSV (blank NPs) have the same anti-inflammatory activity as NPs with RSV, while only in the presence of PF127 the anti-inflammatory activity is enhanced.
Line 346: Before the abbreviation MFI the full words should be added, since this is the first time the abbreviation appears in the text.
Lines 346 - 348: Although the mean value of 1% PF127 in Figure 5b is the lowest, the overlap of error bars shows that this is not statistically significant.
Lines 369 - 370: I can not understand the difference between the terms oral delivery and oral administration that the authors use in the text.
Figure 6 b and c: Although the bars for RSV-NPs and 5%PF127 seem almost the same (particularly if the overlapping of error bars is taken into account) RSV has a p < 0,05 difference from the DSS control, while the 5%PF127 has a p < 0,01 difference from the DSS control.
Line 412: Figure S2a is not provided, so the claim can not be verified.
Line 425 - 426: Figure S2b is not provided, so the claim can not be verified.
Line 429 and 437: Probably “in vivo” is meant instead of simply “vivo”.
References
The references do not have the same morphology and in some of them crucial elements (year of publication, journal, volume or pages) are missing.
Reviewer 2 Report
The paper on mucus penetrating silk nanotherapeutics is nicely written and interesting, even to an outsider of the field. I have no hesitation in recommending publication pending some rather minor amendments, which I have outlined below.
The study is set up in the introduction on the premise that treating ulcerative colitis is of high importance and so intensive efforts need to be made to find safe treatments and son on. However, at no point is a good case for using silk biomaterials ever made. Please see the literature where the case for using silk biomimetics in various industries including medicine has been outlined in extreme detail (e.g. Blamires 'Silk: Exploring Nature's Superfibre', and references within) and build the case.
Please enlarge and embellish the figures, they are currently too small to read and difficult to comprehend without additional details, either by animating on the figures themselves or by adding more explanations to the figure legends.
P 3. Line 15. Micrograms uses the greek symbol 'mu' not 'u'.
P. 8 line 31 'hypotheses will need to be further..'
Round 2
Reviewer 1 Report
The authors have devoted a serious effort in improving the quality of the manuscript. There still a few points that, in my opinion need revision.
Lines 129-133: I still have an objection regarding the quantification of the loaded drug in the NPs. The authors are still talking about dispersion of NPs in DMSO. Since, the drug is entrapped within the matrix of NPs, I can not see how simple dispersion and not complete dissolution will result in quantifying the whole amount of loaded drug.
Lines 272-273: Since PPO is hydrophobic, the outer layer should be hydrophobic as well and not hydrophilic.
Lines 347-349: Still in reference 22 there is no author with last name Zhou.
Lines 366-368: Since the authors agree that there is no statistical significance between results, it can not be claimed that the treatment of NPs with 1%PF-127 presented the lower FITC levels
Author Response
Comments and Suggestions for Authors:
Reviewers1#
The authors have devoted a serious effort in improving the quality of the manuscript. There still a few points that, in my opinion need revision.
Lines 129-133: I still have an objection regarding the quantification of the loaded drug in the NPs. The authors are still talking about dispersion of NPs in DMSO. Since, the drug is entrapped within the matrix of NPs, I cannot see how simple dispersion and not complete dissolution will result in quantifying the whole amount of loaded drug.
Response: Thank you very much for your comments. To quantify drug loading amounts, NPs were dispersed and disrupted in DMSO, and the loaded drugs (resveratrol) were dissolved in DMSO (up to 25 mg/mL). Finally, the obtained solution was transferred into a 96-well plate to measure the absorbance at 320 nm using a multimode plate reader. Therefore, the protocol for drug quantification is pretty simply, and this protocol have been well-established in our group and commonly used in our previous studies (Biomaterials 2022;282:121410; Acta Pharmaceutica Sinica B 2022;12(1):406; Journal of Controlled Release 2020;328:454).
Lines 272-273: Since PPO is hydrophobic, the outer layer should be hydrophobic as well and not hydrophilic.
Response: Thank you for the suggestion. During the fabrication process, fibroin was dissolved in deionized water and subsequently PF-127 was added into the fibroin solution. The hydrophilic segment (PEO) of PF127 tended to stretch outward and prevent the aggregation of NPs in the form of a hydrophilic outer layer, whereas the hydrophobic sections (PPO) of PF127 assembled with the insoluble secondary structures of silk fibroin-based NPs to improve their stability. To clarify this point, we revised the explanation in the revised manuscript (Line 266-270, Page 7).
Lines 347-349: Still in reference 22 there is no author with last name Zhou.
Response: Thank you for the reminding. We have revised it.
Lines 366-368: Since the authors agree that there is no statistical significance between results, it cannot be claimed that the treatment of NPs with 1%PF-127 presented the lower FITC levels
Response: Thank you for the reminding. We have modified the corresponding description in the revised manuscript.